

# Anuran diversity indicates that Caatinga relictual Neotropical forests are more related to the Atlantic Forest than to the Amazon

Deborah P. Castro[1,2,*], João Fabrício M. Rodrigues[3], Maria Juliana Borges-Leite[1], Daniel Cassiano Lima[2] and Diva Maria Borges-Nojosa[1,*]

[1] Programa de Pós-Graduação em Ecologia e Recursos Naturais/Departamento de Biologia, Universidade Federal do Ceará, Fortaleza, Ceará, Brazil
[2] Faculdade de Educação de Itapipoca, Universidade Estadual do Ceará, Itapipoca, Ceará, Brazil
[3] Departamento de Ecologia/Instituto de Ciências Biológicas, Universidade Federal de Goiás, Goiânia, Goiás, Brazil
* These authors contributed equally to this work.

## ABSTRACT

The relationships among the morphoclimatic domains of South America have been a major biogeographical issue of recent years. Palynological, geological and phytogeographical data suggest that the Amazon Forest and the Atlantic Forest were connected during part of the Tertiary and Quaternary periods. This study uses amphibians as model organisms to investigate whether relict northeastern forests are a transition between the Amazon Forest and the Atlantic Forest. We compiled matrices of species composition for four different phytogeographic formations and "Brejos de Altitude," and analyzed them using clustering methods and Cladistic Analysis of Distributions and Endemism. Our results indicate that the anurofauna of these northeastern forest relicts is most similar in composition to the areas of the Atlantic Forest included in this study, and most dissimilar to the Amazon Forest, which leads us to affirm that events of biotic exchange were more frequent within the Atlantic Forest areas.

Corresponding author
Deborah P. Castro,
deborah.praciano@uece.br

## INTRODUCTION

The relationships among the morphoclimatic domains of South America have been a major biogeographical issue of recent years (*Pellegrino et al., 2005*; *Cabanne, Santos & Miyaki, 2007*; *Werneck, 2011*; *Werneck et al., 2011*; *Werneck et al., 2012*; *Sobral-Souza & Lima-Ribeiro, 2017*). Efforts to understand these relationships have identified climatic fluctuations of the Quaternary as the main driver of the current distribution of the main forest vegetation types in the Neotropics (*Haffer, 1969*; *Andrade-Lima, 1982*; *Haffer & Prance, 2002*; *Martins et al., 2009*). Results of palynological (*Ledru et al., 1996*; *De Oliveira, Barreto & Suguio, 1999*; *Bush & Oliveira, 2006*), geological (*De Oliveira, Barreto & Suguio, 1999*), phytogeographical (*Ledru, Salgado-Labouriau & Lorscheitter, 1998*;

*Auler et al., 2004; Santos et al., 2007*) and zoogeographical (*Carvalho, Bortolanza & Soares, 2003; Borges-Nojosa & Caramaschi, 2003; Costa, 2003; Batalha-Filho et al., 2013*) studies suggest that the Amazon Forest and the Atlantic Forest were linked, thereby enabling the exchange of species (*Connor, 1986; Borges-Nojosa & Caramaschi, 2003; Borges-Nojosa, 2007; Carnaval & Bates, 2007; Batalha-Filho et al., 2013*).

The effects of Quaternary climatic fluctuations are evident in northeastern Brazil (*Santos et al., 2007*), which is predominantly comprised of areas of Caatinga but also features forest relicts described by *Andrade-Lima (1966)* and *Veloso, Rangel-Filho & Lima (1991)* as Seasonal Evergreen Forests (Florestas Estacionais Sempre-Verdes). These forests occur on mountains, plateaus and plains of approximately 900 m in elevation, generally located near the coast and whose windward slopes are wetter than the semiarid leeward slopes due to orographic rain (*Araújo et al., 2007*). They were named "Brejos de Altitude" (Relictual Forests) by *Andrade & Lins (1964)* and *Andrade-Lima (1966, 1982)*, and are remnants of vegetation that was probably continuous along the northeast coast of Brazil forming biological corridors between the Amazon Forest and the Atlantic Forest in wetter paleoclimatic periods. They also seem to be refuges within the Caatinga domain for species of these two forest biomes (*Auler & Smart, 2001; Borges-Nojosa & Caramaschi, 2003; Auler et al., 2004*).

The presence of species with disjunct distributions, distributions that encompass the northeastern "Brejos de Altitude" and either the Amazon Forest or the Atlantic Forest (or both), also suggests that the "Brejos-de-Altitude" acted as corridors connecting the two forest biomes (*Andrade-Lima, 1966; Rizzini, 1967; Batalha-Filho et al., 2013*). However, how these connections occurred is not yet fully clear, with two hypotheses currently being used to explain them (*Fiaschi & Pirani, 2009*). According to *Coimbra-Filho & Câmara (1996)*, connections between the Atlantic Forest and the Amazon Forest occurred along the northeastern Atlantic coast, but were destroyed when European settlement began. Based on palynological studies conducted in the Icatu River Valley, Bahia, *De Oliveira, Barreto & Suguio (1999)* found a past plant assemblage that was comprised of species of both the Amazon Forest and the Atlantic Forest, suggesting that a forested ecosystem linking these biomes once existed. This evidence seems to agree with the connection hypothesis suggested by *Oliveira-Filho & Ratter (1995)*, who claimed the existence of a network of riparian forests in Caatinga and Cerrado that were responsible for the links between the Amazon Forest and the Atlantic Forest.

According to *Connor (1986)* and *Carnaval (2002)*, modern distributions of species may be used both to infer local Pleistocene refuges and to test for links among them. In this way, we believe that species with low individual mobility and high philopatry to their natural sites still present a geographical distribution similar to that found in the Pleistocene, and can serve as a model to indicate connections between the Amazon and the Atlantic Forest. Taking this assertion into account, we use amphibians as model organisms for understanding the historical relationship among Brazilian biomes. Populations of amphibians have a tendency to be genetically structured even over short geographical distances and can retain signal of historical events that can be helpful in determining their current distribution (*Zeisset & Beebee, 2008*), reinforcing that modern distributions of
these animals may be used to infer area relationships. This study aims to evaluate the relationships between the "Brejos de Altitude" in northeastern Brazil and the Amazon Forest and the Atlantic Forest using the composition of anuran assemblages. We used anurans as a model to test the hypothesis that "Brejos de Altitude" functioned as important corridors connecting these two forested Brazilian biomes, as suggested by *Andrade-Lima (1982)*. We expected the current anuran fauna of the "Brejos de Altitude" to have a high proportion of species in common with the Amazonian and Atlantic forest formations.

## MATERIAL AND METHODS

### Data Selection

We performed a search for studies with data on species composition of frog assemblages of the northeastern "Brejos de Altitude" and of the three Brazilian phytogeographic formations probably connected in the past: (1) Amazon (Amazon and pre-Amazon region), (2) Atlantic (north, and southeastern/south components) and (3) Cerrado. We also collected anuran composition data from Caatinga (sensu strictu) because it constitutes the landscape matrix in which "Brejos de Altitude" are located. The searches were conducted using ISI Web of Science, SciELO and Google Scholar. We also searched references found in the original articles, publications of the Ministério do Meio Ambiente (MMA) and banks of dissertations and theses. Searches used combinations of the keywords (Species *) and (* diversity and richness) and (* Amazon, Atlantic Forest, Cerrado * * Caatinga, "Brejos-de-altitude" and * anurans *). We reviewed all publications available on the website of the MMA for each phytophisiognomy with assemblages with defined frog species composition. We only selected studies that had a clearly defined study site with known geographical coordinates and known species composition and richness for its frog assemblage. To avoid problems related to taxonomic bias, we excluded from the analysis species that were cited as undetermined in the original publications which included all those registered as "sp" (unidentified species), "gr" (group), "cf" (confers) and "aff" (affinis), according to the criteria proposed by *Araújo, Condez & Sawaya (2009)* and *Forlani et al. (2010)*. To minimize misinterpretation due to the "effect of the taxonomist" we checked synonyms using Amphibian Species of the World (*Frost, 2018*).

### Data analysis

We first constructed a presence-absence matrix where the columns represented species, and the rows represented sites. After we calculated compositional dissimilarities between the study sites using the Sorensen and Simpson indexes, which are commonly used in ecological and biogeographic studies and evaluate different forms of composition change (*Koleff, Gaston & Lennon, 2003*). We used clustering methods to evaluate the relationships between different study sites (*Rosen, 1992*; *Ron, 2000*). We used cladistic analysis of distributions and endemism (CADE) to generate a testable hypothesis of area-relationships between the "Brejos de Altitude" in northeastern Brazil and the Amazon Forest and the Atlantic Forest using the composition of anuran assemblages. CADE is an appropriate

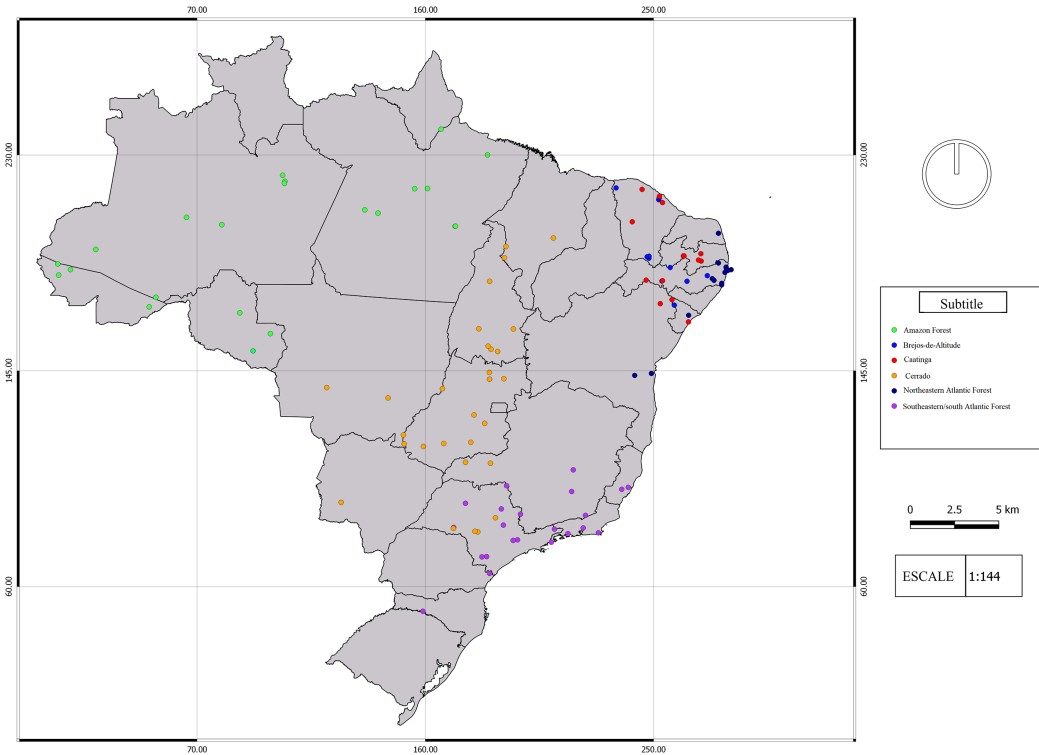

**Figure 1 Map of the study area.** Map of the study sites analyzed in the study. More details of each site may be found in the Supplementary Material. The different vegetation types are represented by colors: southeast/south Atlantic Forest = Purple; northeastern Atlantic Forest = Navy Blue; Caatinga = Red; Cerrado = Orange; Brejos de Altitude = Blue; Amazon = Green. Map Author: Maria Juliana Borges–Leite.

method for examining biogeographical signal, provided hierarchical information is incorporated and sample size is large (*Porzecanski & Cracraft, 2005*; *Nihei, 2006*). CADE was implemented in the software Tree Analysis Using New Technology Version 1.5 (*Goloboff & Catalano, 2016*). In CADE analysis, we included a new "area" containing only zeros in our presence-absence matrix to represent the "outgroup" site. We also included the distribution of the genders and families of the species sampled in our literature search. Tree searches were performed using 1,000 replicates of Wagner trees followed by the Tree Bisection Reconnection swapping algorithm saving 10 trees per replication. We generated a strict consensus tree using the most parsimonious trees found and evaluated the support of this tree using the Bremer support. We grouped study sites by their dissimilarities using an unweighted pair group method using arithmetic averages (UPGMA) algorithm. Second, we ordered areas using non-metric multidimensional scaling (NMDS). We used the statistical software R (*R Development Core Team, 2010*) and the package "Vegan" (*Oksanen et al., 2010*) for the implementation of these analyses.

## RESULTS

We found 93 publications that addressed 113 study sites that met the criteria established for this research. We found a total richness of 423 species of frogs distributed among 11

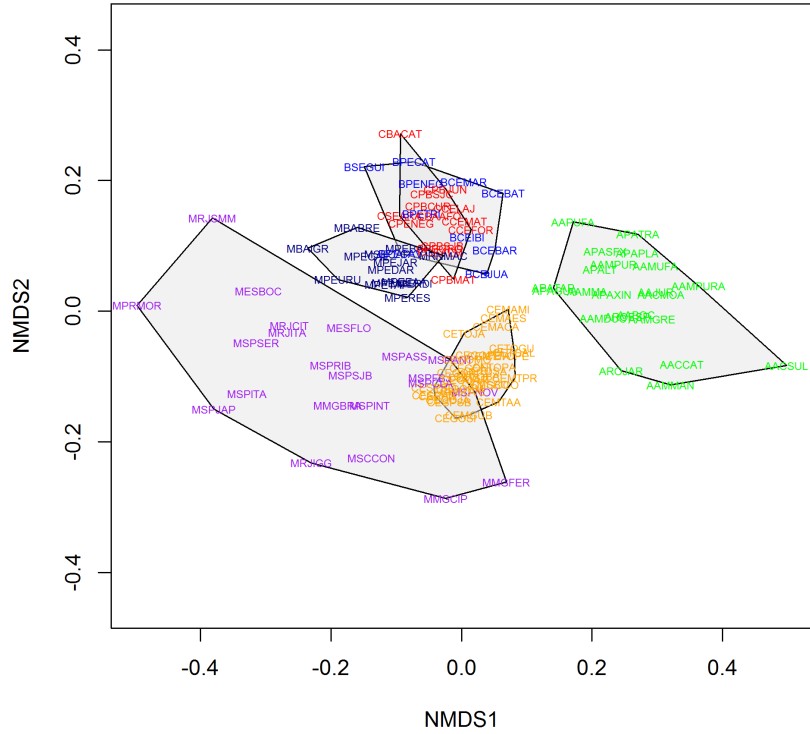

**Figure 2 Non-metric multidimensional scaling (using Simpson index) of the 113 areas analyzed in this study.** Polygons represent the studied vegetation types: southeastern/south Atlantic Forest = Purple; northeastern Atlantic Forest = Navy Blue; Caatinga = Red; Cerrado = Orange; Brejos de Altitude = Blue; Amazon = Green.

locations of "Brejos de Altitude," 22 of Amazon Forest, 14 of Caatinga, 30 of Cerrado and 36 of Atlantic Forest, the latter being divided into northeastern (14) and southeastern/south (22) regions (Fig. 1 and Supplementary Materials). The dendrograms from the UPGMA analyses had similar results, with a slight inconsistency between them. The Simpson's UPGMA showed that Amazon Forest sites are all clustered in a single group and that southeastern/south Atlantic Forest sites are clustered together with Cerrado (Fig. 2). Only two areas of northeastern Atlantic Forest are found within these South–Southern Atlantic Forest groups. In contrast, Sorensen's UPGMA demonstrated that at least four areas of the Southeast/South Atlantic Forest are closely linked to the Amazon Rainforest (Fig. 3). The study sites of "Brejos de Altitude" had species compositions more similar to areas of the northeastern Atlantic Forest and Caatinga, being located with those in the dendrograms. Cophenetic Correlation Coefficients were high (Sorensen = 0,880 and Simpson = 0,838), indicating that the dendrograms are adequately representing real dissimilarities among sites.

Non-metric multidimensional scaling highlighted the existence of the five distinct groups found in UPGMA: (1) Amazon Forest; (2) Cerrado; (3) southeastern Atlantic Forest; (4) northeastern Atlantic Forest and (5) Caatinga, "Brejos de Altitude" (Figs. 4 and 5). The northeastern Atlantic Forest also formed a point of intersection between the southeastern Atlantic Forest and the group of "Brejos de Altitude" and Caatingas. The Cerrado are closely
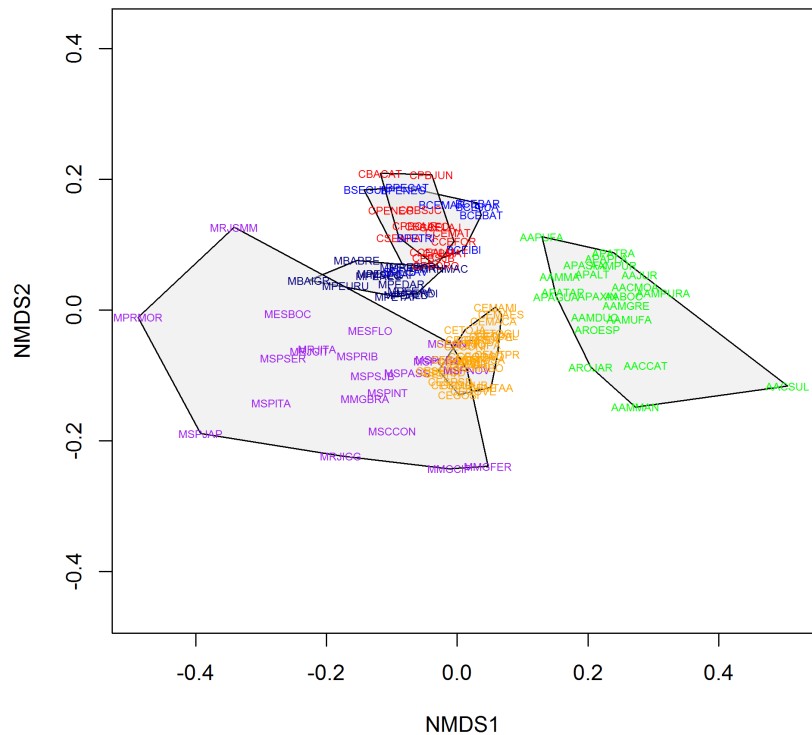

**Figure 3 Non-metric multidimensional scaling (using Sorensen index) of the 113 areas analyzed in this study.** Polygons represent the studied vegetation types: southeastern/south Atlantic Forest = Purple; northeastern Atlantic Forest = Navy Blue; Caatinga = Red; Cerrado = Orange; Brejos de Altitude = Blue; Amazon = Green.

linked to southeastern Atlantic Forest. The stress values of the NMDS analyses were 0.181 (Sorensen index) and 0.189 (Simpson index) for the two dimensions, indicating that there is little distortion in the data and the distances are well supported. We compared the data from these analyses (UPGMA and NMDS) and found that areas of "Brejos de Altitude" had greater similarity with Caatinga and areas of northeastern Atlantic Forest.

The tree search found 100 most parsimonious trees, which had a score of 2,198. The 11 "Brejos de Altitude" were divided into five groups on our consensus cladogram (Fig. 6): (1) Two "Brejos de Altitude" (Serra da Guia-BSEGUI and Brejo dos Cavalos-BPECAV) located in Sergipe and Pernambuco formed a basal polytomy with all others phytogeographic formations; (2) Two areas located in the State of Pernambuco (Triunfo-BPETRI and Buíque-BPECAT); (3)Two areas located in the State of Ceará (Serra de Maranguape-BCEMAR and Ibiapaba-BCEIBI); (4) Three areas located in the State of Ceará (Juazeiro-BCEJUA, Crato-BCECRA and Barbalha-BCEBAR); and (5) one area located in Ceará (Baturité—BCEBAT) and one area of Pernambuco (Serra Negra-BPENEG) formed a single group and were closest to areas of Caatinga.

The CADE generally reinforced the results found in the cluster analysis: (1) the Amazon Forest formed a clade separated from all other vegetation types; (2) the CADE seems to reinforce that the areas of "Brejos de Altitude" are most commonly found closer to the Caatinga and Atlantic Forest Northeast.

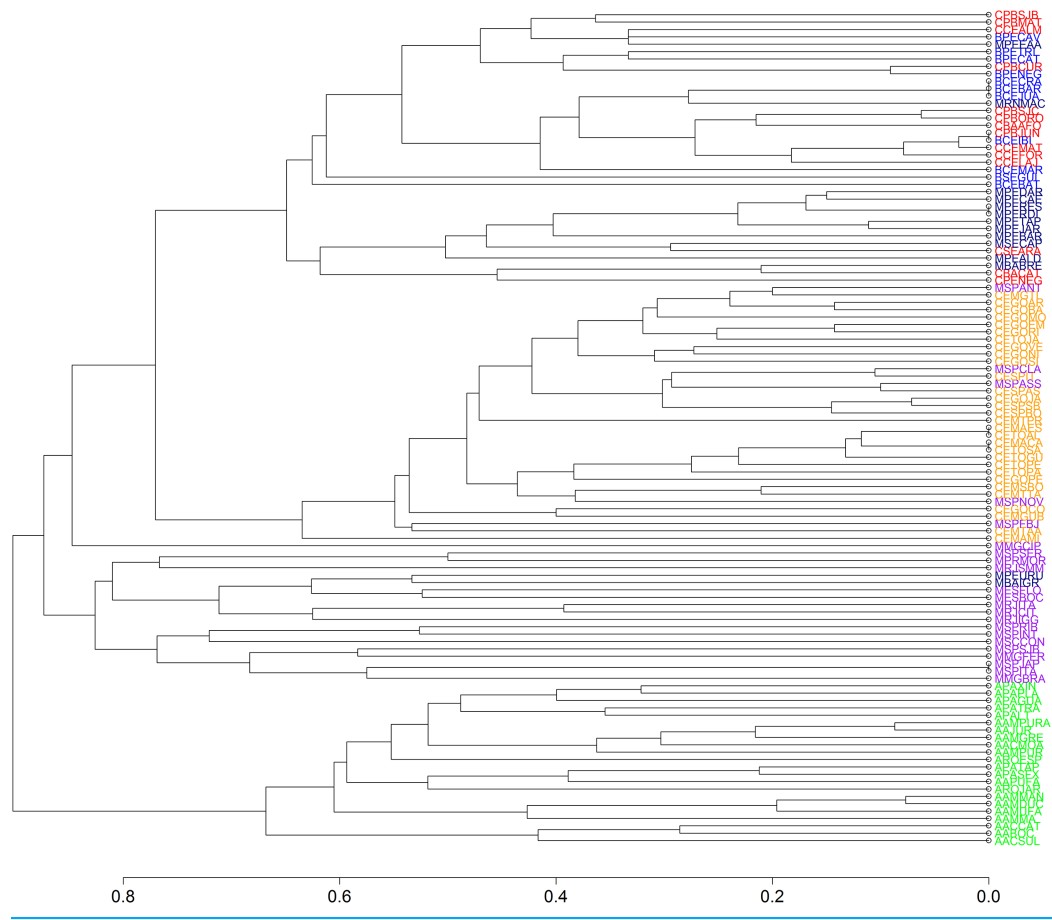

**Figure 4 Dendrogram of similarity between study areas obtained using Unweighted Pair Group Method using Arithmetic averages (UPGMA) and Simpson index for 113 areas included in our study.** The areas are identified by codes found in the Supplementary Materials. The different vegetation types are represented by colors: southeastern/south Atlantic Forest = Purple; northeastern Atlantic Forest = Navy Blue; Caatinga = Red; Cerrado = Orange; Brejos de Altitude = Blue; Amazon = Green.

## DISCUSSION

In our study, we found that anuran assemblages of "Brejos de Altitude" are more similar to areas of Caatinga and Atlantic Forest in northeastern Brazil than to the Amazon Forest. Besides, "Brejos de Altitude" sites are also more similar to distant areas of southeastern/south Atlantic Forest and Cerrado, than to Amazon areas. These results contradict the hypothesis of *Andrade-Lima (1982)* and *Bigarella & Andrade-Lima (1982)* who proposed that the Atlantic Forest and Amazon Forest were interconnected via the "Brejos de Altitude" of northeastern Brazil, and that the current distribution of species resulted from the fragmentation of this ancestral biota.

The great similarity between "Brejos de Altitude" and the northeastern Atlantic Forest was also found by other authors. *Borges-Nojosa & Caramaschi (2003)* and *Borges-Nojosa (2007)*, studied the distribution of reptiles in "Brejos de Altitude" in the state of Ceará, and found greater similarity to areas of Atlantic Forest. *Tabarelli, Silva & Gascon (2004)*
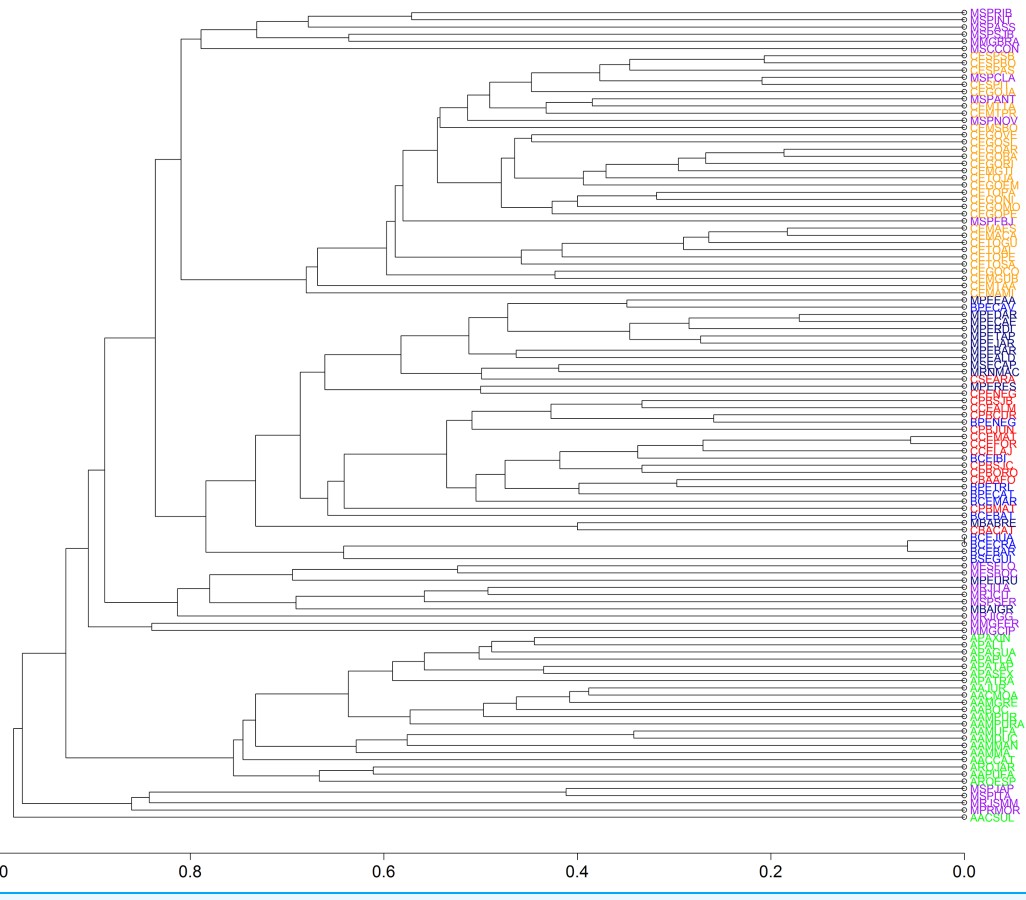

**Figure 5 Dendrogram of similarity between study areas obtained using Unweighted Pair Group Method using Arithmetic averages (UPGMA) andthe Sorensen/Bray-Curtis index for 113 areas included in our study.** The areas are identified by codes found in the Supplementary Materials. The different vegetation types are represented by colors: southeastern/south Atlantic Forest = Purple; northeastern Atlantic Forest = Navy Blue; Caatinga = Red; Cerrado = Orange; Brejos de Altitude = Blue; Amazon = Green.

and *Rodal, Barbosa & Thomas (2008)* found similar results for plants of "Brejos de Altitude." In a phylogeographic study of two species of anurans, *Proceratophrys renalis* (*Miranda-Ribeiro, 1920*) and *Pristimantis* gr. *ramagii* (*Boulenger, 1888*), from "Brejos de Altitude," *Carnaval & Bates (2007)* concluded that these areas are more closely related to the Atlantic Forest than to the Amazon Forest.

The existence of latitudinal differences in the Atlantic Forest, which form a southern component unlike other areas located further north, was also found by some other studies (*Amorim & Pires, 1996*; *Costa et al., 2000*; *Silva, Souza & Castelleti, 2004*; *Carnaval et al., 2009*). *Muller (1973)* distinguished three different sub-centers in the Atlantic Forest (Pernambuco, Bahia and São Paulo). Our results demonstrate that this pattern of differentiation is valid for anurans as well. There is greater similarity between the northeastern component of the Atlantic Forest and areas of "Brejos de Altitude." This similarity may be explained by historical factors and the small geographical distance between the areas of this study.

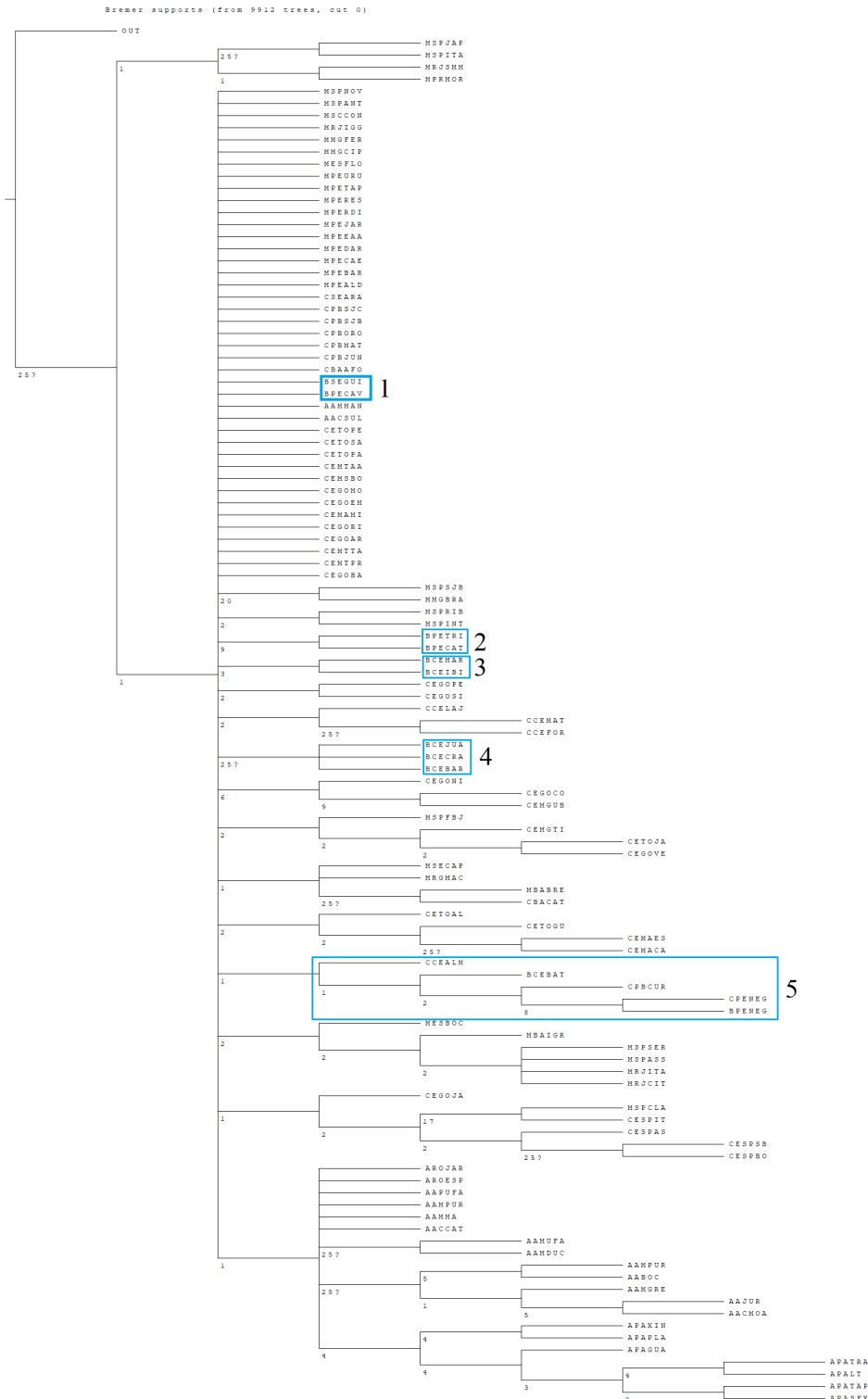

**Figure 6 CADE Bremer support tree consensus.** The strict consensus tree derived from the 100 most parsimonious trees found using the cladistic analysis of distributions and endemism (CADE) for the 113 areas included in our study. Bremer supports are represented in the nodes of the strict consensus cladogram. The values "25?" mean that an increase of more than 25 steps is necessary to dissolve this clade.

We also found a high degree of similarity between "Brejos de Altitude" and Caatinga. According to *Borges-Nojosa & Caramaschi (2003)* the influence of Caatinga on the species composition of the "Brejos de Altitude" is undeniable because this semiarid biome is the matrix in which the relictual forests are inserted. The strong association of "Brejos de Altitude," northeastern Atlantic Forest and Caatinga suggest that "Brejos de Altitude" be considered an ecotone region. Ecotones are transition zones between two adjacent ecological systems. They contain organisms of each of the overlapping communities and organisms that are restrict to the ecotone (*Holland, 1988*; *Risser, 1995*), thereby making them of great interest to conservationists (*Mcarthur & Sanderson, 1999*).

Genetic studies of species found in the humid "Brejos de Altitude" in northeastern Brazil suggest that they do not all have the same evolutionary histories (*Carnaval, 2002*; *Carnaval & Bates, 2007*), and probably resulted from two different former expansions that occurred in the Holocene and Pleistocene (*Borges-Nojosa & Caramaschi, 2003*). In our analysis, areas of "Brejos de Altitude" were not also clustered in a single group, reinforcing these possible multiple origins of these areas. Probably the hypothesis of the "Brejos de Altitude" being a natural biogeographical unit of *Silva & Casteletti (2003)* is not valid. *Carnaval & Bates (2007)* suggest that the "Brejos de Altitude" do not have a common biogeographical origin, and *Vanzolini (1981)* and *Borges-Nojosa & Caramaschi (2003)*, already highlighted that the patterns of distributions of species of amphibians and reptiles show that each currently recognized "Brejo de Altitude" has a unique composition of species.

However, the relationships we found in our study may be specific for anurans. This assertion corroborates that proposed by *Batalha-Filho & Miyaki (2014)*, which states that the processes of diversification within the Brazilian phytogeographic formations may possibly be related to the ecological dependencies and the requirement of habitats inherent to the characteristics of each organism. *Prance (1979*, *1982)*, in a study of plants, and *Teixeira, Nacinovic & Tavares (1986)* with birds, concluded that the Amazon influence on these groups is greater than the Atlantic influence. Anurans, unlike birds and certain plants that can be easily dispersed, have low individual mobility and do not usually disperse over long distances (*Ron, 2000*; *Zeisset & Beebee, 2008*). This may explain the greater influence of the northeastern Atlantic Forest areas on the anuran fauna of the "Brejos de Altitude" since they are geographically closer than the Amazonian areas.

The fact that Sorensen's UPGMA has indicated that some areas of the Southeastern Atlantic Forest have species composition similar to some areas of the Amazon seems to indicate that these areas may share a common history. Probably, the historical contacts between these phytogeographic formations comprise distinct temporal connections (*Batalha-Filho et al., 2013*) through the Cerrado and the Caatinga. On the other hand, despite the evidence found in our study that contradicts a link between the northeastern "Brejos de Altitude" and the Amazon Forest, we cannot be certain that these links have not occurred in the recent past. The disjunct distributions of some taxa indicate that there may have been conditions in the Pleistocene that allowed the expansion of

rainforests (*Andrade-Lima, 1982*; *Pennington, Prado & Pendry, 2000*; *Santos et al., 2007*; *Borges-Nojosa et al., 2017*). Such disjunct distributions have been recorded for plants (*Santos et al., 2007*), birds (*Teixeira, Nacinovic & Tavares, 1986*), mammals (*Costa, 2003*) lizards and amphisbaenids (*Borges-Nojosa & Caramaschi, 2003*). At least two genera of frogs (*Pristimantis* and *Adelophryne*) have such a disjunct distribution, occurring in "Brejos de Altitude" and in other forested areas of Brazil.

## CONCLUSIONS

Our results indicate that the anuran fauna of "Brejos de Altitude" is most similar in composition to areas of the Atlantic Forest and highly dissimilar to areas of the Amazon Forest and Cerrado. In addition, the northeastern "Brejos de Altitude" seem to have different histories and have a strong ecotonal character, which reinforces the importance of conservation of these formations. To better understand the current distribution of taxa of these relictual forests and how historical and geological processes have shaped their distribution, more inventories and phylogeographic studies of frog species for the "Brejos de Altitude" and Caatinga are needed.

## ACKNOWLEDGEMENTS

We thank Geraldo Jorge Barbosa de Moura, Marinus Steven Hoogmoed, Mirco Solé and Paulo Cascon for provided valuable comments which helped us to improve the manuscript. We are grateful to David James Harris and Eric Wild, who revised the English version of this manuscript.

### Funding

This study was supported by the Conselho Nacional de Desenvolvimento Científico e Tecnológico (CNPQ) (PVE 401800/2013-0) and Coordenação de Aperfeiçoamento de Pessoal de Nível Superior (CAPES). The funders had no role in study design, data collection and analysis, decision to publish, or preparation of the manuscript.

### Grant Disclosures

The following grant information was disclosed by the authors:
Conselho Nacional de Desenvolvimento Científico e Tecnológico: PVE 401800/2013-0.
Coordenação de Aperfeiçoamento de Pessoal de Nível Superior.

### Competing Interests

The authors declare there are no competing interests.

### Author Contributions

- Deborah P. Castro conceived and designed the experiments, performed the experiments, analyzed the data, contributed reagents/materials/analysis tools, prepared figures and/or tables, authored or reviewed drafts of the paper, approved the final draft.

- João Fabrício M. Rodrigues conceived and designed the experiments, performed the experiments, analyzed the data, contributed reagents/materials/analysis tools, prepared figures and/or tables, authored or reviewed drafts of the paper, approved the final draft.
- Maria Juliana Borges-Leite conceived and designed the experiments, performed the experiments, contributed reagents/materials/analysis tools, authored or reviewed drafts of the paper, approved the final draft.
- Daniel Cassiano Lima conceived and designed the experiments, performed the experiments, contributed reagents/materials/analysis tools, authored or reviewed drafts of the paper, approved the final draft.
- Diva Maria Borges-Nojosa conceived and designed the experiments, performed the experiments, contributed reagents/materials/analysis tools, authored or reviewed drafts of the paper, approved the final draft.

## Data Availability

The raw data are provided in the Supplemental Files.

## Supplemental Information

Supplemental information for this article can be found online at http://dx.doi.org/10.7717/peerj.6208#supplemental-information.

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
