# Peer review of "Anuran diversity indicates that Caatinga relictual Neotropical forests are more related to the Atlantic Forest than to the Amazon"

_PeerJ, doi:10.7717/peerj.6208_

## Round 0.1 · original submission · Major Revisions

Both reviewers suggested that your paper needs improvements before publication. They listed some issues that you should address in the new version. One of the reviewers recommended that you should use a modern method of biogeographical analyses such as CADE as well as include Cerrado as a region of interest. I believe you should take this suggestion into account in the new version.

I also recommend that you get editing help from someone with full professional proficiency in English to ensure your text is free of any language problem.

When resubmitting your manuscript, please carefully consider ALL points mentioned in the reviewers' comments, explain every change made, and provide proper rebuttals for any remarks not addressed.

Reviewer 1 ·

Basic reporting

The subject of the manuscript is interesting and important for the comprehension of the past connections between the Amazon and the Atlantic Forest through Northwestern Brazil. Nevertheless, I bring some considerations for manuscript improvement, considering that it can be a good contribution to the field.

The instrumental use of the English is good, but somehow it could be improved a little more, prioritizing fast reading. Actually, it is not an instrumental English issue, but a scientific writing issue, trying to present an overview of the subject, a gap of knowledge, and your contribution to the field in a more objective way.

Objectives are not well exposed in the introduction. The first five paragraphs position the reader to the climatic fluctuations of the Pleistocene and their effects on the Northwestern Brazil’s vegetation dynamics. They also show some studies supporting mesic vegetations inside the Caatinga (“brejos de altitude”) as a “footprint” of an ancient connection between the Amazon and the Atlantic Forest, as exposed in the text. Nevertheless, it does not bring the author to the objective of the work, which seems to test the hypothesis of “brejos de altitude” acting as this “footprint” of a past connection between the Amazon and the Atlantic Forest (or assessing how each biome – Amazon and Atlantic Forest – contributed to the formation of this connection, providing species that were sampled on the “brejos de altitude”, if these contributions were similar or not). Actually, I didn’t find a hypothesis in the introduction. In my opinion, these five paragraphs could also be reorganized in just three, with the same amount of information. Doing so, you must have more room to explore your objective, hypotheses, and approaches to assess your objectives.

Still about these first paragraphs, the literature used for background to the subject is classical (Haffer, 1969; Vanzolini & Williams, 1970; Vanzolini, 1981; Andrade-Lima, 1982; Oliveira-Filho & Haffer, 1995; Haffer & Prance, 2002; Costa, 2003; Auler, 2004; Santos et al. 2007) and used precisely. Nevertheless, there are more recent studies that contributed to the state of art of the field and should be included in. There are some in terms of phylogeography and ecological niche modelling:

PRATES, I. et al. A mid-Pleistocene rainforest corridor enabled synchronous invasions of the Atlantic Forest by Amazonian anole lizards. Molecular Ecology, v. 25, n. 20, p. 5174-5186, 2016. ISSN 1365-294X. Disponível em: < http://dx.doi.org/10.1111/mec.13821 >.

SOBRAL-SOUZA, T.; LIMA-RIBEIRO, M. S.; SOLFERINI, V. N. Biogeography of Neotropical rainforests: past connections between Amazon and Atlantic Forest detected by ecological niche modeling. Evolutionary Ecology, v. 29, n. 5, p. 643-655, 2015/07/03 2015. ISSN 0269-7653. Disponível em: < http://dx.doi.org/10.1007/s10682-015-9780-9 >.

LEDO, R. M. D.; COLLI, G. R. The historical connections between the Amazon and the Atlantic Forest revisited. Journal of Biogeography, v. 44, n. 11, p. 2551-2563, 2017. Disponível em: < https://onlinelibrary.wiley.com/doi/abs/10.1111/jbi.13049 >.

COSTA, G. C. et al. Biome stability in South America over the last 30 kyr: Inferences from long‐term vegetation dynamics and habitat modelling. Global Ecology and Biogeography, v. 27, n. 3, p. 285-297, 2018. Disponível em: < https://onlinelibrary.wiley.com/doi/abs/10.1111/geb.12694 >.

The Introduction’s last paragraph tries to justify the use of modern distribution of species to answer questions that deals with the past (ancient connections between the Amazon and Atlantic Forest) and also tries to justify the use of amphibians in this approach. In my opinion, more literature should be incorporated in this last paragraph, in order to give more support to the approach of using modern distribution of species to assess questions of the past. Apparently, you included more references to justify the use of amphibians for your objectives than to justify the approach of using current species distribution data to assess past phenomena. Therefore, more literature should be included in order to support your approach.

Still about the last paragraph. Please, expose the problematic to be assessed by your study. Please, try to answer these questions as well: why is your study a good contribution to the state of the field of ancient connections between the Amazon and the Atlantic Forest? In what level your study contributes to the knowledge about past connections between the Amazon and the Atlantic Forest through the “brejos de altitude”?

I have also noticed that you cited unpublished data and personal communications. I recommend you to cut them out of the manuscript, since these studies were not submitted to peer review for publication and we are not sure of the validity of these conclusions.

This paragraph is just a suggestion. The title is a little bit vague and not consonant to the conclusions of your work. I think they should be changed to something more direct, such as “Anuran diversity indicates that Caatinga relictual Neotropical forests are more related to the Atlantic Forest than to the Amazon”. Faster (19 words) and tell all the story. Please, take it just as a suggestion.

This paragraph is just a suggestion. Considering the purposes of this journal and some other authors proposing more reproducible studies, would you like to include the original species matrix you used in your analyses as a supplemental material? Please, take it just as a suggestion, considering reproducibility criteria.

Figure 1 to 4. Please include in the captions a mention to the supplementary material.

Experimental design

The study is an original primary research due to its approach, even not using primary data, but data from literature. Nevertheless, the research question is not well defined and should be included in. It also does not state how the research fills an identified knowledge gap. Nevertheless, the subject is relevant and meaningful.

I have noticed that you excluded some taxonomic inconsistencies prior to analysis. However, why do you use species with “cf” in the analysis? In my opinion, It seems as imprecise as categories such as aff, gr, and sp. I think you should show in the text a reason for that.

I have noticed that you included biotas from the Amazon, Atlantic Forest, Brejos de Altitude and Caatinga into your analyses. However, why did you not include Cerrado into your analyses? There are some “brejos” that contain a significant amount of Cerrado species on it (e.g. Crato), supporting the inclusion of some Cerrado localities in the analyses. Moreover, hypotheses of pluvial forests expansions and retractions through the Pleistocene are also coupled with expansions and retractions of other biomes nearby (Cerrado and Caatinga as well).

Moreover, why did you not try to use a stricter biogeographical analysis, such as Cladistic Analysis of Distribution and Endemism (CADE)? The use UPGMA is interesting, but it is not considered an “biogeographical” analysis per se. Please, provide in the test a reason you would rather UPGMA than a CADE. Notice that a study cited in your introduction (Costa et al 2007) uses PAE and CADE for the same purpose you did. In my opinion, you had better use a CADE with branch support (e.g. Bremer) instead of dendrograms, but if you have a good reason for using dendrograms, please provide in the text.

Validity of the findings

The results show some novelty not assessed before. The discussion presented is interesting. Maybe it could include some sentences or a paragraph more when including Cerrado biota into analysis.

Additional comments

The study is interesting and has potential for publication. Nevertheless, there are some issues that need to be tackled.

Reviewer 2 ·

Basic reporting

OK. The authors should avoid the use "personal communications" and unpublished data. They did not test hypothesis, they use cluster analysis.

Experimental design

The main concern about the study is that the methods the authors used do allow infer “process” on the relation of “Brejos” with Amazon, Atlantic Forest and Caatinga, but only “patterns”. The used cluster and ordination analyses, which could be nice to reveal patterns, but not to infer process. To do this, the need to use phylogenetic/phylogeographic information.

Validity of the findings

The findings were ok to reveal patterns, but they should avoid use them to infer process.

Additional comments

Dear author,

First, I would like to thank the opportunity to contribute with your manuscript. The manuscript untitled “What does the diversity of anurans have to say about relationships between a relictual Neotropical forest and the Amazon and Atlantic forests?” provide an interesting information about the relationships about these areas. However, there are several points that should be addressed prior the publication. I did several comments on the pdf file, which I think should be very important for the manuscript. The main concern about the study is that the methods the authors used do allow infer “process” on the relation of “Brejos” with Amazon, Atlantic Forest and Caatinga, but only “patterns”. Again, thanks for the opportunity, and I hope that my suggestions help the authors in their manuscript.

Annotated reviews are not available for download in order to protect the identity of reviewers who chose to remain anonymous.

---

## Round 0.2 · Major Revisions

Deborah,

Thanks for your revision. However, as you can see in the referee's letter, you have not followed all critical suggestions outlined in the first revision. Reviewers asked you to use a formal biogeographic method to show the relationships among areas. I agree with the reviewer's comments as they are well supported by the literature.

Perhaps you can use both cluster and PAE (or CADE) analyses and discuss the similarities and differences between them while considering the assumptions of both methods.

I hope to see an updated version of your paper soon. Following the best practices adopted at PeerJ, I will send the new version to reviewers again.

Reviewer 1 ·

Basic reporting

Overall, the written text was well improved if compared to the first draft. It was improved in textual logic, what improved its structure and quality. The objective was also clearer compared to the first draft, which also improved the quality of the discussion section.

Fig. 1 has a green point out of map range.

The only criticism I still have to this study is applied to the statistical analysis. After analyzing the first draft, we suggested the use of a stricter biogeographical analysis (CADE) instead of clustering analyses, because clustering approaches cannot be used to infer biogeographical processes, but only patterns (even criticized, CADE creates groups based on shared presences and common ancestry, whereas ordination approaches create groups based only on overall similarity). In other words, even criticized among some biogeographers, CADE is a better alternative than clustering analyses. Moreover, the papers cited by the authors (Morrone 2014; Nihei 2006) do not prohibit the use of PAE/CADE methods, but recommend the use of it in specific cases (such as in exploratory contexts or in dynamic approaches). What should be avoided is the indiscriminate use of the analysis.

The authors also justify the use of clustering approaches based on a recent study (Barroso 2016), applied to prosobranch gastropods. Nevertheless, this study is clearly more applicable to a community structure level (beta diversity, and not to historical biogeography), inferring patterns of species distribution rather than processes. As this study is anchored in a historical issue (ancient connections between the Amazon and the Atlantic Forest through “brejos”), the use of clustering approaches will clearly impoverish your discussion and conclusions.

In short, a CADE is not the most appropriate approach for some biogeography studies (and it was used extensively used since in its inception. That is the real problem). Nevertheless it is certainly a better alternative than a cluster analysis, because the former creates groups based on shared presences and common ancestry, whereas clustering approaches create groups based only on overall similarity, including noise to your analysis and not inferring process, just patterns. Therefore, the support for the use of a cluster analysis by the authors is weak. Obviously, CADE was just a suggestion. Other biogeographic methods could be used as well (i.e. Endemicity Analysis, considered to perform better than PAE; or even other methods). Another alternative is to make both analysis together (cluster and CADE) and compare the results.

Considering that the most important point of my suggestions was not properly considered, unfortunately I am not able to recommend the study for approval, although I noticed very significant improvements from the first version to this one.

Experimental design

already mentioned in "basic reporting".

Validity of the findings

no comment.

Additional comments

Considering that the most important point of my suggestions was not properly considered, unfortunately I am not able to recommend the study for approval, although I noticed very significant improvements from the first version to this one.

---

## Round 0.3 · accepted · Accept

Congratulations. Please review your paper again to make sure all references follow the same style when working with our production team.

#